# Single-catalyst high-weight% hydrogen storage in an *N*-heterocycle synthesized from lignin hydrogenolysis products and ammonia

Daniel Forberg[1], Tobias Schwob[1], Muhammad Zaheer[1,2], Martin Friedrich[1], Nobuyoshi Miyajima[3] & Rhett Kempe[1]

Large-scale energy storage and the utilization of biomass as a sustainable carbon source are global challenges of this century. The reversible storage of hydrogen covalently bound in chemical compounds is a particularly promising energy storage technology. For this, compounds that can be sustainably synthesized and that permit high-weight% hydrogen storage would be highly desirable. Herein, we report that catalytically modified lignin, an indigestible, abundantly available and hitherto barely used biomass, can be harnessed to reversibly store hydrogen. A novel reusable bimetallic catalyst has been developed, which is able to hydrogenate and dehydrogenate *N*-heterocycles most efficiently. Furthermore, a particular *N*-heterocycle has been identified that can be synthesized catalytically in one step from the main lignin hydrogenolysis product and ammonia, and in which the new bimetallic catalyst allows multiple cycles of high-weight% hydrogen storage.

[1] Lehrstuhl für Anorganische Chemie II–Katalysatordesign, Universität Bayreuth, 95440 Bayreuth, Germany. [2] Department of Chemistry, SBA School of Science and Engineering, Lahore University of Management Sciences (LUMS), Lahore 54792, Pakistan. [3] Bayerisches Geoinstitut, Universität Bayreuth, 95440 Bayreuth, Germany. Correspondence and requests for materials should be addressed to R.K. (email: kempe@uni-bayreuth.de).

The search for sustainable living under conditions in which there is an increasing energy demand calls for the replacement of the dominant fossil fuel-based technologies that are currently employed. Two central issues of this global challenge are energy storage and the use of indigestible and abundantly available biomass as a carbon source. The concept of a 'hydrogen economy' including hydrogen-based energy storage was suggested over 40 years ago[1]. Hydrogen possesses a high energy density, is an environmentally friendly energy carrier and is suitable for mobile application[2,3]. Due to significant technical and safety concerns regarding cryogenic liquid and compressed hydrogen, chemical hydrogen storage is a highly attractive alternative especially for large-scale energy storage[2,4]. Molecular hydrogen carriers based on the diversity of molecular compounds from which hydrogen can be liberated are particularly promising. $H_2$-release has been described for a variety of different classes of compounds, such as cycloalkanes, N-heterocycles, 1,2-BN-heterocycles, methanol or formic acid, ammonia borane, hydrous hydrazine and hydrazine borane[5]. Methanol might be the most attractive or mature molecular hydrogen carrier for irreversible hydrogen storage. It can be produced sustainably from $CO_2$ (ref. 6) and hydrogen liberation can occur efficiently under mild conditions[7,8]. In this low temperature reforming process, three equivalents of dihydrogen are liberated, with one originating from water, the oxygen atom of which contributes to $CO_2$ formation[9]. For the reversible storage of hydrogen, N-heterocycles have been earmarked as promising candidates[10]. The presence of N atoms in carbocyclic compounds allows dehydrogenation to take place at lower temperatures when compared with the corresponding cycloalkanes[11,12]. Exhaustive hydrogenation and acceptor-less dehydrogenation by the very same catalyst is a domain of homogeneous catalysis and has been reported for 2,6-dimethylpyridine[13] and 2,6-dimethyl-1,5-naphthyridine[14]. Interestingly, the latter, having a maximum hydrogen storage capacity of 6.0 wt%, has been multiply hydrogenated and dehydrogenated using a molecular Ir catalyst. Reversible single-catalyst hydrogen storage based on catalytic peptide formation has been demonstrated very recently[15,16]. An important issue with regard to any hydrogen storage material/compound is its sustainable production and scalability[11]. We recently disclosed a sustainable catalytic synthesis concept in which a combination of condensation and hydrogen liberation steps allows the synthesis of aromatic N-heterocycles from different alcohols[17]. The synthesis of pyridines[18,19] proceeds with the liberation of three equivalents of $H_2$. Pyrimidines can be synthesized from alcohols and ammonia generating up to four equiv. of $H_2$ (ref. 20). Based on this concept of alcohol-to-heterocycle conversion, it seems feasible to develop a reaction that converts key building blocks of the overabundant and indigestible biomass lignin to N-heterocycles suitable for reversible high-wt% hydrogen storage.

Herein, we report on the development of a robust reusable bimetallic catalyst able to completely hydrogenate and dehydrogenate N-heterocycles in a most efficient way. In addition, we also disclose a catalytic one-step synthesis of octahydrophenazine starting from the main lignin hydrogenolysis product[21,22] and ammonia. The novel reusable catalyst mediates reversible high-wt% hydrogen storage and depletion in this sustainably synthesized N-heterocycle.

## Results

### Catalyst synthesis and characterization.
First, we became interested in developing an efficient reusable catalyst capable of completely hydrogenating and dehydrogenating N-heterocycles under relatively mild conditions. N-ethylcarbazole (NEC)[10], which has a theoretical hydrogen storage capacity of 5.8 wt%, was chosen as a model N-heterocycle for catalyst identification. It is a solid with a melting point of 68 °C and a negligible toxicity, whereas the hydrogenated product, dodecahydro-N-ethylcarbazole (12H-NEC), is a liquid under ambient conditions with a boiling point above 220 °C. Thus, it can be handled similarly to diesel fuel[23]. Consequently, large-scale automated and especially decentralized energy applications for domestic or commercial buildings seem feasible by means of NEC and an appropriate catalyst[23,24]. The hydrogenation of NEC can be accomplished by heterogeneous ruthenium (Ru) catalysts[10,25] while palladium (Pd)-based solid catalysts have been reported to be efficient in the dehydrogenation of NEC[26]. Reusable single catalysts would extend the applicability of NEC-based hydrogen storage significantly. In their patents, Pez et al.[10,27] performed the hydrogenation and dehydrogenation of NEC in a single reactor system using two different catalysts. Given, that the most efficient catalysts for NEC-hydrogenation are Ru based and the most efficient ones for 12H-NEC-dehydrogenation are Pd based, potential novel single-catalysts for reversible hydrogen storage should comprise both metals. Synergistic effects of Pd catalysts, when combined with other transition metals, have been reported for hydrogenation reactions[28] and can be rationalized by assuming a lower hydride-binding energy in a metal alloy[29]. Recently, the silicon carbonitride (SiCN) matrix has been introduced as an attractive support for metal catalysts[30,31]. Amorphous SiCN is very robust thermally and chemically inert[32]. The N-atoms of such a support permit the generation of very small metal nanoparticles despite the high pyrolysis temperature and high metal loadings[33]. The polymer based synthesis protocol for SiCN, operating by crosslinking polysilazanes and subsequent pyrolysis, allows the introduction of porosity in various ways. For instance, metal-mediated nanoporosity[34] and meso-structuring via sacrificial polyolefin components have been reported[35–38].

By mixing the commercially available Ru complex 1 and the aminopyridinato palladium complex 2 (ref. 39) with the commercially available polysilazane HTT1800, followed by crosslinking and pyrolysis, bimetallic nanocomposite catalysts are accessible (Fig. 1a). The variation of the molar ratio of both metals allows an optimization of the performance of the catalyst with regard to both, the hydrogenation and dehydrogenation steps. A Pd-to-Ru ratio of 2 was found to be optimal ($Pd_2Ru@SiCN$). $N_2$ sorption revealed metal mediated porosity with a bimodal pore size distribution centered at 1.5 (minor) and 4.6 nm (major) and a specific surface area (Brunauer-Emmett-Teller model) of 82 $m^2 g^{-1}$ for $Pd_2Ru@SiCN$. Transmission electron microscopy (TEM) revealed a nanocomposite in which the metal particles are homogenously distributed over the SiCN matrix (Fig. 1b). High-resolution TEM analysis resulted in a d-spacing of 224.1 ± 2.1 pm, which is in accordance with the expected value of 224.6 pm for the (111) reflex of cubic crystalline Pd (Fig. 1b, inset). The averaged Pd particle size was determined to be 1.6 nm by TEM (Fig. 1c). High-angle annular dark-field and energy dispersive X-ray spectroscopy (EDX) investigations confirmed the existence of metallic Pd nanoparticles and of homogenously distributed Ru clusters smaller than 1 nm in size and undetectable via X-ray diffraction (Fig. 1d–g).

### Catalyst screening and reversible hydrogen storage.
A comparison of $Pd_2Ru@SiCN$ with a variety of Pd, Ru and Ir catalysts including catalyst mixtures reveals the superiority of our bimetallic catalyst in hydrogen uptake and release (Fig. 2a). The $Pd_2Ru@SiCN$ catalyst hydrogenated NEC quantitatively at 110 °C and only 20 bar $H_2$ pressure and thus exceeded the efficiency of other commercial Ru catalysts and of Ir@SiCN

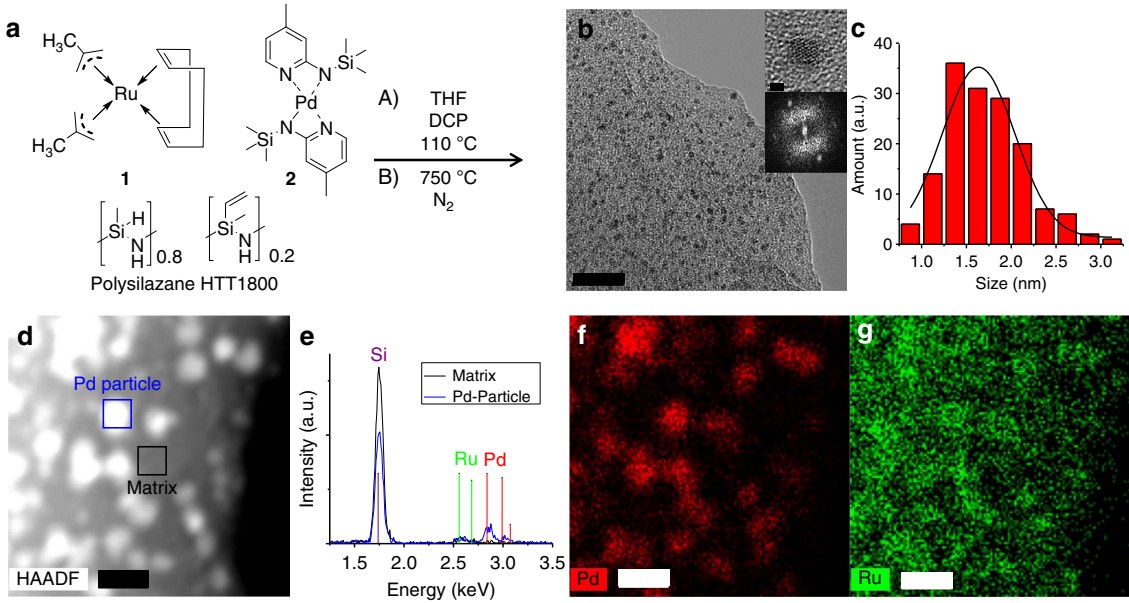

**Figure 1 | Synthesis and characterization of the nanocomposite catalyst Pd₂Ru@SiCN.** (**a**) A solution of complex **1** and complex **2** in tetrahydrofuran (THF) was mixed with the polysilazane HTT1800 and crosslinked at 110 °C for 24 h using dicumylperoxide (DCP) as a radical initiator. The solvent was removed and the resulting black–brown metallopolymer was pyrolysed under $N_2$ atmosphere at 750 °C. (**b**) High-resolution TEM picture of the catalyst (scale bar, 20 nm) with magnification of one Pd nanoparticle and fast Fourier transform (FFT, scale bar, 1 nm). (**c**) Pd particle size distribution. (**d**) High-angle annular dark-field (HAADF) picture (scale bar, 5 nm). (**e**) Energy dispersive X-ray spectroscopy (EDX) analysis of the matrix and Pd particle. (**f**) EDX mapping of Pd (scale bar, 5 nm). (**g**) EDX mapping of Ru (scale bar, 5 nm). a.u., arbitrary units.

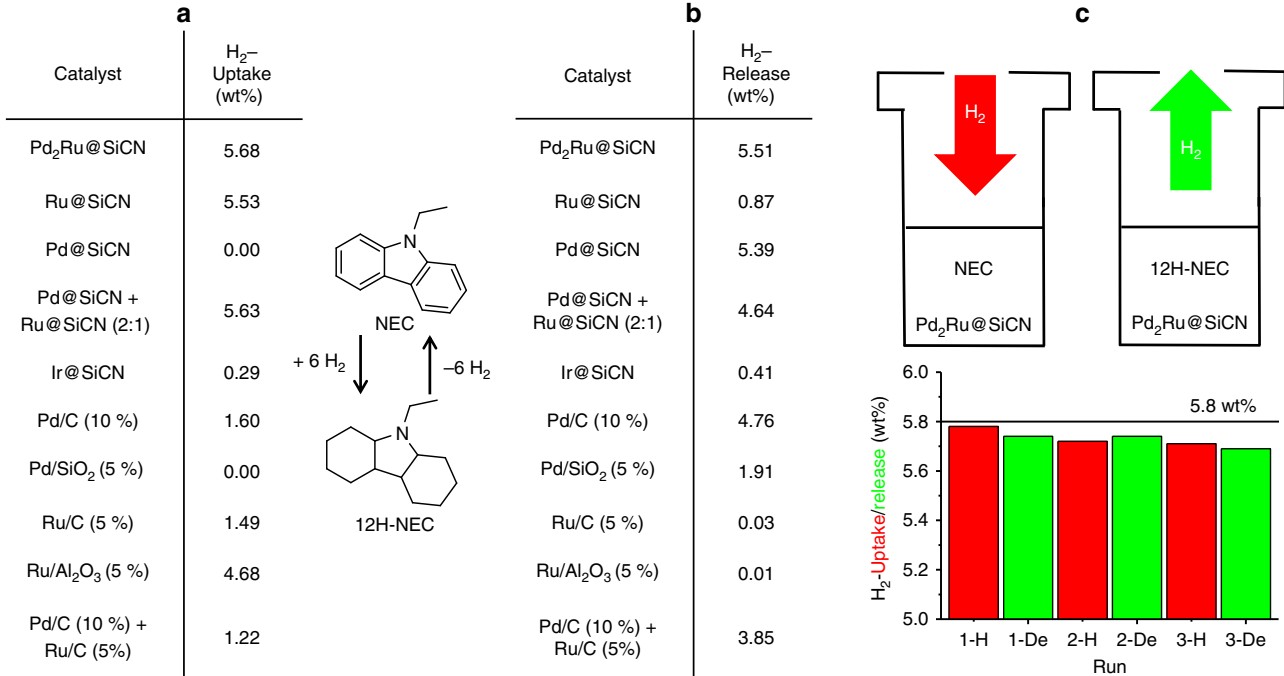

**Figure 2 | Catalyst screening and reversible hydrogen storage in NEC.** (**a**) Hydrogenation of NEC to 12H-NEC; 110 °C, $p(H_2) = 20$ bar, 1 mmol NEC, 0.52 mol% active metal, 36 h. (**b**) Dehydrogenation of 12H-NEC to NEC; 180 °C, 2 mmol 12H-NEC, 0.52 mol% active metal, 7 h. The hydrogen uptake and release values were calculated by gas chromatography (GC) and GC-mass spectrometry (MS) analysis. The maximum hydrogen uptake or release is 5.8 wt% (calculated based on the substrate). (**c**) Catalyst reusability and reversible hydrogen storage in NEC: 1.0 g (5.12 mmol) N-ethylcarbazole, 200 mg Pd₂Ru@SiCN (0.52 mol% active metal); hydrogenation: 110 °C, 20 bar $H_2$, 36 h; dehydrogenation: 180 °C, 20 h. The hydrogen uptake and release values were calculated by GC and GC-MS analysis and based on the preceding step.

by far. We included an efficient heterogeneous Ir catalyst[40] in our comparison due to the reported perhydrogenation and perdehydrogenation activity of a homogenous Ir catalyst[14].

The exhaustive dehydrogenation of 12H-NEC to NEC by the Pd₂Ru@SiCN catalyst could be performed at 180 °C reaction temperature within 7 h. Again, other commercially

available catalysts are less efficient. The monometallic Ru- or Pd@SiCN catalysts mediate either the hydrogenation or the dehydrogenation, respectively, whereas a combination of both steps is not feasible for the one or the other catalyst. The Pd$_2$Ru@SiCN catalyst is superior also to a 2:1 mixture of commercially available Pd and Ru catalysts, as well as to a mixture of the corresponding SiCN catalysts. The two metals in one catalyst approach (Pd$_2$Ru@SiCN) results in more efficiency in both steps especially the hydrogen release, less catalyst mass and a higher specific surface area (porosity) as the single metal hydrogen release catalyst Pd@SiCN. Further results of the catalyst screening experiments are shown in Supplementary Tables 1–3. The reusability of the Pd$_2$Ru@SiCN catalyst was verified by three consecutive hydrogen storage cycles with NEC (Fig. 2c). NEC was hydrogenated to 12H-NEC at 110 °C and 20 bar H$_2$ pressure within a hydrogen storage capacity of > 5.7 wt%. Subsequently, the reaction vial was heated to 180 °C in an oil bath whereupon 5.7 wt% of hydrogen were released within 20 h. This procedure was repeated three times without any significant loss of catalytic activity or hydrogen storage capacity. For a detailed product distribution, please see Supplementary Table 4. The wt% of hydrogen has been calculated based on the N-heterocycle conversion. For one release experiment, the liberated hydrogen gas was collected and quantified, as well as analysed for purity by gas chromatography (GC). Calculated (5.16 wt%) and collected (4.93 wt%) hydrogen are in good agreement. No CO impurity could be detected.

**Development of a novel hydrogen storage system.** The storage of hydrogen with the NEC/12H-NEC system is limited to a capacity of 5.8 wt% (The current DOE target is 5.5 wt% for the year 2017. For details, please visit their website. Targets for onboard automotive hydrogen storage systems, US Department of Energy, Office of Energy. Efficiency and Renewable Energy, http://energy.gov/eere/fuelcells/hydrogen-storage). Now, with an

efficient catalysts system for reversible hydrogen storage in hand, we set out to identify an N-heterocycle with a higher storage capacity and amenable to a simple, sustainable synthesis. Lignin, a three-dimensional biopolymer, composes 15–30% of the lignocellulosic biomass, which is abundant along with negligible food chain competition[41]. In addition, it is barely used. The main lignin building blocks are 1,2-dialkoxybenzenes (Fig. 3, green circles) and their hydrogenolysis proceeds mostly in combination with hydrogenation[21] leading to cyclohexane-1,2-diol[22]. Cyclohexane-1,2-diol could be reacted with ammonia[42] applying the borrowing hydrogen[43] methodology leading to a 2-aminocyclohexanol intermediate, which then can undergo dehydrogenation and condensation steps[17,40,44] to form perhydrophenazine(s) (Fig. 3a). Ir@SiCN converts cyclohexane-1,2-diol and ammonia selectively to 1,2,3,4,6,7,8,9-octahydrophenazine in 74% isolated yield (Fig. 3a). Octahydrophenazine can be quantitatively dehydrogenated to phenazine with the Pd$_2$Ru@SiCN catalyst and then used in consecutive hydrogen storage cycles (Fig. 3b) or used directly. Hydrogenation was performed by dissolving phenazine in a dioxane/water mixture at 115 °C and 50 bar H$_2$ pressure. The addition of water led to higher hydrogenation rates. After removal of the solvent, the residual tetradecahydrophenazine was dehydrogenated at 190 °C. The hydrogen uptake/release was demonstrated for seven consecutive cycles under identical conditions. For a detailed product distribution, please see Supplementary Table 5. Besides some minor variations of hydrogen uptake and/or release over these seven cycles, the Pd$_2$Ru@SiCN catalyst was still capable of mediating a hydrogen uptake of more than 7.0 wt% in the last cycle (Fig. 3b). The released hydrogen was collected, quantified and analysed regarding its purity for one experiment. The calculated released hydrogen based on the conversion of the N-heterocycle (6.26 wt%) was in good agreement with the actually liberated and collected amount (6.19 wt%). We could not detect CO formation.

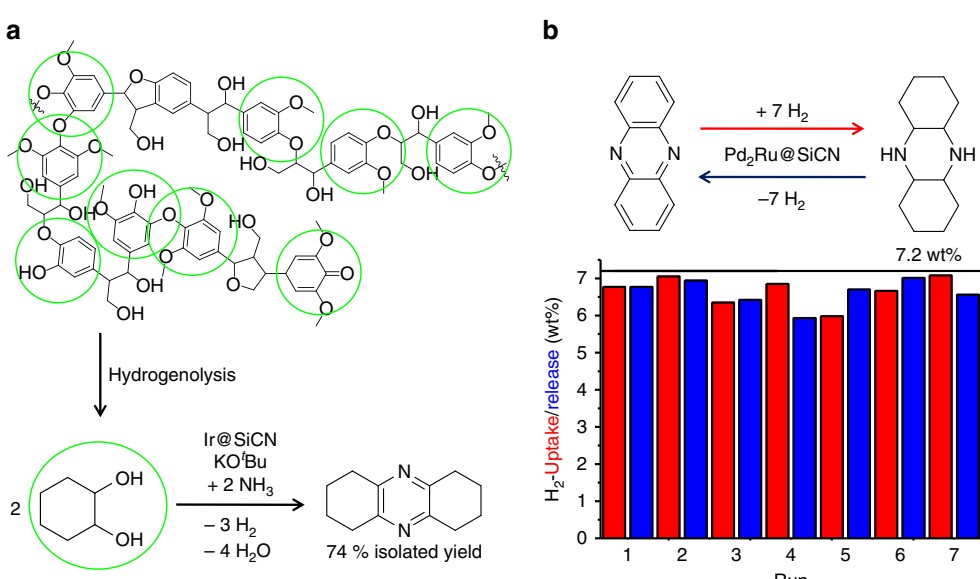

**Figure 3 | Sustainable synthesis of octahydrophenazine from cyclohexanediol and ammonia and its use as a reversible hydrogen carrier.**
(**a**) A section of the lignin structure with alkoxybenzene subunits marked by green circles (top). Lignin can be converted to cyclohexane-1,2-diol using existing methodologies (hydrogenolysis). Synthesis of octahydrophenazine from cyclohexane-1,2-diol and ammonia (bottom). (**b**) Catalyst reusability and reversible hydrogen storage with phenazine: 360 mg (2 mmol) phenazine, 70 mg Pd$_2$Ru@SiCN (0.36 mol% active metal). Hydrogenation: 115 °C, 50 bar H$_2$, 2 ml dioxane, 0.5 ml water, 24 h. Dehydrogenation: 190 °C, 0.75 ml digylme, 24 h. The hydrogen uptake and release values were calculated by GC and GC-MS analysis and based on the preceding step. The maximum hydrogen uptake or release is 7.2 wt% (calculated based on the substrate not considering the solvent). The values for the uptake and release of H$_2$ are based on those of the preceding step.

## Discussion

In summary, we demonstrated that lignin, an overabundant, indigestible and barely used type of biomass, can be catalytically modified to *N*-heterocyclic compounds suitable for reversible high-wt% hydrogen storage. A novel robust, reusable, and highly efficient bimetallic catalyst for the exhaustive hydrogenation and dehydrogenation of such *N*-heterocycles was also developed to complete a conceptionally new lignin based energy storage system. We expect it to initiate an intensified quest by scientists for similar integrative energy storage systems, that can be produced and operated sustainably and in accordance with conservation-of-elements strategies. The new bimetallic hydrogenation–dehydrogenation catalyst is interesting in its own right and we expect it to find ample application to sustainable organic synthesis.

## Methods

**Catalyst synthesis.** Under an argon atmosphere, complex **1** (29 mg, 0.09 mmol), complex **2** (84 mg, 0.18 mmol), and dicumylperoxide (5 mg, 2.9 wt%) were dissolved in 1.5 ml of tetrahydrofuran and the commercially available polysilazane HTT1800 (173 mg) was added. The reaction vial was placed in a preheated oil bath at 120 °C for 24 h. After removing of the solvent by reduced pressure, the brown-black solid was pyrolysed under $N_2$ atmosphere: heating of 1 °C per minute to 300 °C, holding time 1 h at 300 °C, heating of 5 °C per minute to 750 °C, holding time 1 h at 750 °C and cooling of 4 °C per minute to room temperature. For a detailed characterization of the different SiCN catalysts please see Supplementary Figs 1–11. For more details, please see also Supplementary Methods.

**Reversible hydrogen storage.** A reaction vial containing a solution of 360 mg (2 mmol) phenazine and 70 mg Pd$_2$Ru@SiCN in 2 ml dioxane and 0.5 ml water was given in a 250 ml stainless steel autoclave. The autoclave was flushed with hydrogen three times and a hydrogen pressure of 50 bar was adjusted. The reactor was heated to 115 °C for 24 h. After cooling to room temperature, a sample for GC and GC-mass spectrometry was taken and the mixture was transferred to a 20 ml Schlenk tube. The solvents were removed under reduced pressure and 0.75 ml diglyme was added. The tube was evacuated and flushed with argon for three times and a slight argon flow of 4–6 ml min$^{-1}$ was adjusted. The tube was placed in a preheated metal bath at 200 °C (190 °C reaction temperature). After 24 h the mixture was cooled to room temperature under argon atmosphere and a sample for GC and GC-mass spectrometry analysis was taken. The released gas was additionally analysed by GC and no CO formation could be detected (Supplementary Figs 12–13). The devices used for the reversible hydrogen storage are shown in Supplementary Fig. 14. For more details, please see also Supplementary Methods.

**Synthesis of octahydrophenazine.** In a glass vial, Ir@SiCN (ref. 40) catalyst (250 mg, 0.08 mol% active metal) and cyclohexane-1,2-diol (2.32 g, 20 mmol) were dissolved in 12 ml of degassed water. The vial was placed in a 250 ml stainless steel autoclave and was flushed three times with ammonia. An ammonia pressure of 5 bar was adjusted and the reaction mixture was stirred at 105 °C. After 24 h the ammonia atmosphere was released and the reactor was again pressured with 5 bar ammonia and the reaction mixture was allowed to stir for another 24 h. The reactor was cooled to room temperature with an ice bath and the mixture was extracted two times with 100 ml diethylether. The organic phases were combined and were extracted with 150 ml of 0.1 M HCl. The water phase was neutralized with 2 M NaOH and extracted two times with 100 ml diethylether. The combined organic phases were dried over Na$_2$SO$_4$ and the solvent was removed under reduced pressure giving the light yellow product in 74% yield. For a spectroscopic analysis, please see Supplementary Figs 15–20 and Supplementary Note 1. For more details, please see also Supplementary Methods.

**Data availability.** All data is available from the authors on reasonable request.

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

## Acknowledgements

We acknowledge financial support from the Deutsche Forschungsgemeinschaft, SFB 840, B1.

## Authors contributions

D.F. carried out the synthesis and hydrogen storage experiments and analysed the spectroscopic data. T.S. carried out the hydrogen release quantification experiments. M.Z. synthesized the Pd@SiCN catalyst. M.F. and N.M. carried out the TEM and energy dispersive X-ray spectroscopy characterizations. D.F. and R.K. designed the experiments and co-wrote the manuscript.

## Additional information

**Competing financial interests:** The authors declare no competing financial interests.

