## [Peer review file · Nature Communications]

Reviewers' comments:

Reviewer #1 (Remarks to the Author):

The paper is very interesting. The heterogeneous catalyst Pd₂Ru@SiCN is novel and useful. It effectively catalyzes both hydrogenation/dehydrogenation reactions of N-ethylcarbazole; previously separate Ru and Pd catalysts were required for these reactions. In addition, a new hydrogen storage system based on phenazine, originating from biomass-derived cyclohexanediol, was developed using this catalyst.

I recommend publication in Nat. Comm. However, there are some significant points that should be addressed prior to publication:

(1) It is not clear to me if the produced hydrogen gas was collected, or just calculated based on conversion of substrate. The generated hydrogen must be collected and quantified in the best experiments. It should also be analyzed (by GC and/or IR) to make sure that CO is not produced (this is an important point regarding use in fuel cells).

(2) As described regarding the phenazine system, solvent is used in the dehydrogenation (and hydrogenation) reactions. This obviously reduces the wt% of hydrogen produced. I guess that the reported wt% of hydrogen is calculated based on the substrate, not taking into consideration the solvent. This should be mentioned in the footnotes of the Tables in the text and SI.

(3) The activity of the bimetallic catalyst Pd₂Ru@SiCN is compared to that of Pd and Ru supported on various supports, but not on SiCN. To be directly relevant, comparison to Pd@SiCN and to Ru@SiCN, as well to as a mixture of both, is needed. Otherwise the synergistic bimetallic effect is in doubt. I realize that this would require significant more work, unless the authors have already prepared these supported mono-metallic compounds. Hence it should be left to the authors to decide if they are ready to do it. Otherwise, this issue should be mentioned in the text.

(4) Introduction, p. 2: review on acceptorless dehydrogenation: C. Gunanathan, D. Milstein Science, 2013, 341, 1229712

Reviewer #2 (Remarks to the Author):

This paper describes the reversible hydrogen storage using phenazine as an organic hydride molecule. A single heterogeneous catalyst composed of palladium and ruthenium exhibited high activity for both dehydrogenation and hydrogenation. It should be also noted that the substrate phenazine can be catalytically synthesized from ammonia with 1,2-cyclohexanediol, which is easily available by hydrogenolysis of lignin.

It must be highly important to develop an efficient method for hydrogen storage. The results shown in this work provides a new promising system for hydrogen storage based on "organic hydride". The content of this paper seems to have sufficient novelty and urgency suitable for the publication in high-quality journal. This paper is concisely written, and the results are clearly indicated in the main text and supporting information. References are appropriately cited. Therefore, this referee recommends this paper to be published in Nature Communications. However, the authors should address to the following comments, and revise the manuscript appropriately.

1. The ratio of Pd : Ru in the catalyst should be further studied. Only the results with Pd : Ru = 2:1, 1:1, and 1:2 have been studied. Activity of the catalyst with Pd : Ru = 4 :1 or the catalyst including only Pd should be examined.

2. Characterization including spectroscopic analysis of 14H-Phen (tetradecahydrophenazine) should be performed.
3. In the dehydrogenation of 14H-Phen, measurement of volume of evolved hydrogen gas should be performed (only determination of the yield of organic products is insufficient). Additionally, purity of evolved hydrogen gas should be analyzed.
4. In the hydrogenation of phenazine, the reaction was carried out in the presence of a small amount of water (0.5 mL). The reason why water was added should be explained.
5. In the supporting information, illustration or photo of the apparatus used for dehydrogenation and hydrogenation should be added.

Point-by-point response to the reviewer's comments

Reviewer #1:

"The paper is very interesting. The heterogeneous catalyst Pd₂Ru@SiCN is novel and useful. It effectively catalyzes both hydrogenation/dehydrogenation reactions of N-ethylcarbazole; previously separate Ru and Pd catalysts were required for these reactions. In addition, a new hydrogen storage system based on phenazine, originating from biomass-derived cyclohexanediol, was developed using this catalyst.

I recommend publication in Nat. Comm. However, there are some significant points that should be addressed prior to publication:"

"Our response": Thanks to Reviewer #1 for evaluating and improving our manuscript!

"(1) It is not clear to me if the produced hydrogen gas was collected, or just calculated based on conversion of substrate. The generated hydrogen must be collected and quantified in the best experiments. It should also be analyzed (by GC and/or IR) to make sure that CO is not produced (this is an important point regarding use in fuel cells)."

"Our response": For NEC, we collected the released hydrogen and could see a very good agreement. We now collected and quantified the generated hydrogen for both N heterocycles. In addition, we checked for impurities especially for CO.

"Our alteration": We added a brief discussion to the manuscript explaining the experiments and the outcome. In addition, we described the method used to quantify in detail in the SI (Supplementary Information).

"(2) As described regarding the phenazine system, solvent is used in the dehydrogenation (and hydrogenation) reactions. This obviously reduces the wt% of hydrogen produced. I guess that the reported wt% of hydrogen is calculated based on the substrate, not taking into consideration the solvent. This should be mentioned in the footnotes of the Tables in the text and SI."

"Our alteration": We mentioned in footnotes of the manuscript text and the SI that the wt% of hydrogen is based on the substrate and not considering the solvent if solvent(s) were used.

"(3) The activity of the bimetallic catalyst Pd₂Ru@SiCN is compared to that of Pd and Ru supported on various supports, but not on SiCN. To be directly relevant, comparison to Pd@SiCN and to Ru@SiCN, as well to as a mixture of both, is needed. Otherwise the synergistic bimetallic effect is in doubt. I realize that this would require significant more work, unless the authors have already prepared these supported mono-metallic compounds. Hence it should be left to the authors to decide if they are ready to do it. Otherwise, this issue should be mentioned in the text."

“Our response”: We synthesized the pure metal catalysts and included them in the comparison. They are able to mediate the one step efficiently but not the other one. Ru@SiCN mediates hydrogenation, unfortunately, it is nearly inactive in dehydrogenation and Pd mediates dehydrogenation without significant rates in hydrogenation. Synergism results from a few facts: (1.) the metal based porosity is an outcome of the presence of Ru. By incorporating Ru, the “Pd catalyst” becomes more porous. (2.) The mixture of the two catalyst is less efficient especially in hydrogen release and has (3) significantly more catalyst mass.

“Our alteration”: We included the catalyst comparison into the manuscript (Figure 2) and added a brief discussion. We also added the catalyst synthesis and characterization to the SI

“(4) Introduction, p. 2: review on acceptorless dehydrogenation: C. Gunanathan, D. Milstein Science, 2013, 341, 1229712”

“Our alteration”: We added the citation of this very interesting review.

Reviewer #2:

“This paper describes the reversible hydrogen storage using phenazine as an organic hydride molecule. A single heterogeneous catalyst composed of palladium and ruthenium exhibited high activity for both dehydrogenation and hydrogenation. It should be also noted that the substrate phenazine can be catalytically synthesized from ammonia with 1,2-cyclohexanediol, which is easily available by hydrogenolysis of lignin.

It must be highly important to develop an efficient method for hydrogen storage. The results shown in this work provides a new promising system for hydrogen storage based on "organic hydride". The content of this paper seems to have sufficient novelty and urgency suitable for the publication in high-quality journal. This paper is concisely written, and the results are clearly indicated in the main text and supporting information. References are appropriately cited.

Therefore, this referee recommends this paper to be published in Nature Communications.

However, the authors should address to the following comments, and revise the manuscript appropriately.”

“Our response”: Thanks to Reviewer #2 for evaluating and improving our manuscript!

“1. The ratio of Pd : Ru in the catalyst should be further studied. Only the results with Pd : Ru = 2:1, 1:1, and 1:2 have been studied. Activity of the catalyst with Pd : Ru = 4 :1 or the catalyst including only Pd should be examined.”

“Our response”: The request is very similar to the third one of Reviewer #1 and, thus, we respond analogously. For details, please see above.

“Our alteration”: Please see above.

"2. Characterization including spectroscopic analysis of 14H-Phen (tetradecahydrophenazine) should be performed."

"Our response": We performed it and added it.

"Our alteration": Details of the characterization including figures of the ^1H and ^{13}C NMR spectra were added to the SI.

"3. In the dehydrogenation of 14H-Phen, measurement of volume of evolved hydrogen gas should be performed (only determination of the yield of organic products is insufficient). Additionally, purity of evolved hydrogen gas should be analyzed."

"Our response": The request is very similar to the first one of Reviewer #1 and, thus, we respond analogously. For details, please see above.

"Our alteration": Please see above.

"4. In the hydrogenation of phenazine, the reaction was carried out in the presence of a small amount of water (0.5 mL). The reason why water was added should be explained."

"Our response": We see a better hydrogenation rate in water.

"Our alteration": We explained why we use it in one additional sentence in the manuscript.

"5. In the supporting information, illustration or photo of the apparatus used for dehydrogenation and hydrogenation should be added."

"Our alteration": We added a photo as requested to the SI.

REVIEWERS' COMMENTS:

Reviewer #1 (Remarks to the Author):

All my concerns have been satisfactorily addressed in the revised manuscript, which now includes new experiments and the requested data. I recommend publication of this fine paper as is.

Reviewer #2 (Remarks to the Author):

The author has improved the manuscript according to the comments by the referees. I think that the revised version seems to be acceptable for publication in Nature Communications.